# Verbal, Figural, and Arithmetic Fluency of Children with Cochlear Implants

**DOI:** 10.3390/bs13050349

**Published:** 2023-04-22

**Authors:** Renata Skrbic, Vojislava Bugarski-Ignjatovic, Zoran Komazec, Mila Veselinovic

**Affiliations:** 1Faculty of Medicine, University of Novi Sad, 21 137 Novi Sad, Serbia; 2Clinic for Neurology, University Clinical Center of Vojvodina, 21 137 Novi Sad, Serbia; 3Clinic for Otorhinolaryngology and Head and Neck Surgery, University Clinical Center of Vojvodina, 21 137 Novi Sad, Serbia

**Keywords:** deafness, hearing impairment, cognitive functions, phonemic fluency, semantic fluency, non-verbal fluency, mathematical fluency

## Abstract

Cochlear implantation gives children with prelingual severe hearing loss and deafness the opportunity to develop their hearing abilities, speech, language, cognitive abilities and academic skills with adequate rehabilitation. The aim of the research was to analyze verbal, figural and arithmetic fluency and their interrelationship in children with a cochlear implant (CI) and children with normal hearing (NH). A total of 46 children with CI and 110 children with NH, aged 9 to 16, participated in the research. Verbal fluency was assessed using phonemic and semantic fluency, and non-verbal fluency using figural fluency. Arithmetic fluency was assessed using simple arithmetic tasks within the number range up to 100. The results showed that children with CI achieved poorer results in phonemic fluency (z = −4.92; *p* < 0.001), semantic fluency (z = −3.89; *p* < 0.001), figural fluency (z = −3.07; *p* = 0.002), and arithmetic fluency (z = −4.27; *p* < 0.001). In both groups, a positive correlation was obtained between the measured modalities and types of fluency. In the group of children with CI, a sex difference was obtained on the phonemic fluency test, in favor of girls. The age of children with CI was correlated with arithmetic fluency. Verbal, figural and arithmetic fluency of children with CI speak in favor of the importance of early auditory and language experiences.

## 1. Introduction

### 1.1. The Impact of Early Auditory Deprivation and Cochlear Implantation on Development and Functioning in Children

Auditory abilities are one of the important factors of development, and auditory deprivation in early childhood can lead to difficulties in functioning and delays in various aspects of development. A cochlear implant (CI), gives children with prelingual deafness the opportunity for certain auditory experiences, on the basis of which, with adequate and long-term rehabilitation processes, they can develop speech-language [1,2] and cognitive abilities, participate in communication and socialization, learn and be educated in formal and informal contexts, attend regular schools, work and engage in all aspects of daily functioning [3,4]. Thanks to the improvement of the technical characteristics of the implant and the possibility of bilateral implantation, together with the application of adequate rehabilitation and support to the child and family, today’s cochlear implantation has multiple goals. In addition to achieving speech intelligibility without lip reading, using the phone and listening in noisy conditions, today’s goals include the improvement of a person’s social, emotional, and cognitive abilities [4]. However, auditory stimulation with CI is not quite the same as in typical listening conditions. The signal is degraded, and its processing by the auditory pathways and centers is often altered [1]. Nevertheless, even in such circumstances, the brain’s neuroplasticity enables new auditory experiences to be integrated into a person’s experiential fund and to contribute to the further development of abilities and skills in various areas of functioning, especially in the case of early implantation [5]. Although significant progress is being made in the area of speech-language development [6], where research indicates a wide range of achievements, in other areas of functioning different findings have been reported. The achievements of children with CI lag behind their hearing peers in the domain of cognitive and social functioning [7,8] and academic skills [6,9,10]. Nevertheless, there are authors who state that achievements of children with CI are comparable to those of their hearing peers, particularly in the case of early implantation. Khan et al. [11] have found no difference in figural cognitive functions of children with CI and of hearing children, while other authors indicate the significance of early exposure to and acquisition of sign language, as a protective factor in neurocognitive development [12].

The reasons for such a wide range of achievements can be partly explained by factors related to the deafness itself, such as the age of the child at the moment of implantation and activation of the CI, the hearing age of the child, i.e., how much time has passed since the activation of the CI, intellectual abilities, rehabilitation procedures, early language exposure, family and many other factors [2,12,13,14,15,16,17]. In recent times, there are more and more studies that speak in favor of cortical reorganization (cross-modal and intermodal) and missing critical/sensitive periods for the development of certain cognitive abilities, as an additional explanation for the differences in CI outcomes and the lag of children with CI compared to their peers [1]. Kral et al. [18] analyzed the “connectome model of deafness” and showed that understanding the development of CI users requires observing the functioning of the entire brain, i.e., the entire neural network, including other sensory, motor and cognitive systems. In addition to the hypothesis of early auditory deprivation as the cause of difficulties in cognitive functioning and of different neurocognitive organization, some authors propose the hypothesis of early language deprivation, which many CI users experience [19,20]. Over 90% of deaf people have hearing parents [21], and especially in environments with less developed health systems for detection, diagnosis and CI implantation, an oral approach to rehabilitation may be taken; early language stimulation and acquisition may be absent, which also leads to a different neural organization and different modalities of functioning in different domains, such as the cognitive and social [19,20].

In the last decade, executive functions (EFs) of children with CI have increasingly been the focus of research, which emphasizes the importance of early auditory and language experiences for their development. Neurocognitive development in conditions of early auditory deprivation versus actively listening with the help of CI lead to the creation of different (specific) types of neural organization, and therefore, processing patterns, leading to a further different development of abilities and functions which are based on listening [22]. Conway et al. [23] consider early auditory experiences to be particularly important for the experience of temporal patterns, and therefore, for sequential processing, which is based on the maintenance of attention and serial memory. They are the basis of executive functions, such as controlled attention, planning, working memory, fluency and efficiency [24]. The achievements of children with CI in the area of most executive functions lag significantly behind the achievements of hearing peers, particularly in the level of development of verbal working memory, processing speed and fluency, and in the area of attention [24,25,26].

### 1.2. Fluency

The term fluency refers to the ability to perform certain actions or skills quickly and accurately, automatically, with as little effort as possible [27]. In the field of cognitive abilities and processing, fluency is most often associated with EF, either as its component or through correlation with general or specific EF domains, such as working memory, inhibition, and cognitive flexibility [28,29]. Fluency, in this sense, can be verbal and non-verbal, depending on the type of tasks and requirements to be met. Although verbal and non-verbal tasks imply different modalities of processing and response, and thus the involvement of different, distant parts of the CNS, they share common correlates and cognitive processes to a certain extent, especially in the area of the left dorsolateral prefrontal cortex [30]. Fluency assessment tests examine the ability to generate verbal or non-verbal responses according to established criteria and rules within a specified time frame. Their execution requires planning, organized search and performance monitoring, and they are, for this reason, considered a measure of executive functions [31].

#### 1.2.1. Verbal Fluency

Verbal fluency (VF) in adults, as the ability to recall and produce words associated with a previously determined category (semantic fluency—SF) or that begin with a specific sound (phonemic fluency—PF), is linked to the left inferior frontal gyrus, the anterior cingulate cortex and the upper frontal sulcus [32,33]. Phonemic fluency tasks lead to greater activation of the left inferior frontal gyrus, while categorical fluency is more strongly associated with the left fusiform gyrus and left middle frontal gyrus [33]. Newer research supports different activation of brain structures during development, and in the research of Gonzalez et al. [34] in children aged 7 to 13 years, better achievement on verbal fluency tasks was associated with the higher right superior longitudinal fasciculus/arcuate fasciculus, and PF was moderately associated with the lower left superior longitudinal fasciculus/arcuate fasciculus. Verbal fluency is partially related to EF and vocabulary factors, as well as to social, cognitive and health phenotypes [29]. It is influenced by both genetic and environmental factors. The authors believe that VF is best viewed as an “amalgamation” of the overall variance of EF, but is also associated with specific domains of EF, such as working memory and shifting.

#### 1.2.2. Figural Fluency

Figural fluency (FF) is a measure of non-verbal fluency, which implies the ability to generate unique and original nonverbal responses. For the successful performance of FF tasks, adequate visuospatial, graphomotor and motor planning abilities are required, along with maintaining attention and focus on the task, cognitive monitoring and self-monitoring [35]. The complexity of the request leads to activation of the dorsolateral prefrontal cortex, right posterior parietal cortex, prefrontal cortex, and right frontal lobe [30]. Figural fluency can be tested using various tasks, structured and unstructured, such as drawing different versions of figures, connecting dots and completing drawings. [36].

#### 1.2.3. Arithmetic Fluency

A small number of studies in the area of mathematical skills speak in favor of the poorer achievements of children with CI compared to hearing children [37,38], and note an absence of effect of long-term use of CI compared to deaf children who are not users of CI [3]. Like many others, the area of mathematical skills of children with CI requires additional attention, as knowledge from other sciences, especially STEM (science, technology, engineering and math), various professions and the needs of everyday life rely on basic mathematical knowledge, which is thus linked to success and quality of life [39].

In education, fluency is a basic component of reading, writing, arithmetic or proficiency in another language [40]. Achieving fluency is a stage of learning academic skills, based on which the children will be able to apply the learned skill in other contexts, i.e., generalize it and be able to use it flexibly in different situations [27]. By automating, e.g., reading or numeracy skills, cognitive resources are freed up, and the child can focus on more complex demands, such as reading comprehension or solving equations [40]. Furthermore, fluency is used as a measure of the acquisition of academic knowledge, and you can often find terms such as reading fluency, writing fluency, mathematical (arithmetic) fluency in the literature. These measures are often cited as predictors of later academic achievement and/or the need for additional educational support [41].

The term arithmetic fluency (AF) broadens the definition of arithmetic skills to include speed and flexibility in selecting and applying calculation procedures and strategies and the automatic recall of basic numerical facts (declarative knowledge) to arrive at the correct solution. Knowing numerical facts and recalling them easily and quickly from long-term memory allows reducing the cognitive load and freeing up cognitive resources when performing more complex tasks [42]. Greater activity of the left intraparietal sulcus is associated with more successful solving of arithmetic fluency tasks [43].

By the end of the third grade of primary school, students are expected to be able to automatically recall basic arithmetic facts using whole numbers, in all four arithmetic operations [44]. Students who are slow and have to methodically, step by step, approach solving problems using the simple arithmetic operations and the relationship between numbers, require more time to complete the tasks. Over time, as mathematical skills become more complex, the gap in relation to other students increases, and new concepts take more time and become more difficult to adopt. The authors see the reasons for this as relating to working memory. Geary [42], on the other hand, relates difficulties in AF to semantic memory, and Jordan [45] finds a poor sense of number and difficulties in understanding the relationship between numbers as contributory causes of poorer achievements. Other authors link AF to domains of EF such as working memory, inhibition, and shifting [46].

Investigations of different modalities of fluency in children with CI have mostly been undertaken as part of larger cognitive measurements and, as far as we know, have not yet been linked to arithmetic fluency. Most research points to poorer achievements of children with CI on verbal fluency tasks, while the results are different for figural fluency. Although there are studies in the literature that indicate that the achievements of children with CI are comparable to the achievements of children with preserved hearing when assessing mathematical skills, most authors state that users of CI lag behind and have worse achievements. Accordingly, we asked the following questions:What are the achievements of children with CI on tasks of verbal, figural and arithmetic fluency?To what extent are verbal and figural fluency related to the ability to respond quickly and accurately to calculation tasks in children with CI?

The aim of the paper is to examine whether there is a difference between the achievements of children with CI and their hearing peers in the area of verbal, figural and arithmetic fluency and to determine whether the investigated forms of fluency are statistically related to each other.

## 2. Materials and Methods

### 2.1. Subject and Procedures

This cross-sectional study included 156 children, divided into two groups. The first group consisted of 46 children with a cochlear implant (CI), and the second group consisted of 110 children with preserved hearing (normal hearing—NH).

The common criteria for inclusion in the research were: signed consent of the parents and the child, age of the child being between 9 and 16 years, the Serbian language being the native language or the child’s first language, and average non-verbal intellectual abilities.

The criteria for inclusion in the study group included: prelingual profound hearing loss and deafness (>90 dB HL in the better ear which occurred before the third year of age), implanted cochlear implant before the age of 6, active use of the cochlear implant for 5 years and more. Exclusion criteria from the study group: presence of additional developmental disabilities, neurological or psychiatric disorders.

The control group consisted of children with preserved hearing, of average intellectual abilities, who successfully passed hearing screening (otoacoustic emissions). Criteria for the exclusion of children with NH from the control group: children on individualization measures or an individual education program, the presence of developmental disabilities, neurological or psychiatric disorders.

Prior to conducting the research, consent was obtained from the administrations of the institutions where the research was conducted, as well as from their Ethics Committees. The consent of the Ethics Committee of the Faculty of Medicine was also obtained (protocol number 01-39-50/2015).

The research was conducted in the period between June 2016 and June 2017, in two healthcare and five educational institutions in Novi Sad, Belgrade and the surrounding area in Republic of Serbia.

Prior to conducting the testing, written consent was obtained from all parents and children who participated in the research. All parents of children with CI and children with CI who were offered participation in the research gave their written consent. Socio-demographic and socio-economic data, as well as characteristics of the child related to deafness and cochlear implant placement, were obtained from parents and medical records. Based on the consent of parents and children, 49 children with CI were initially included in the research. One child was excluded from the research, since, based on the medical documentation and the psychologist’s findings, he did not meet the criteria for the absence of additional developmental disabilities. With two children with CI, due to problems with attention and concentration, further examination was ceased after the administration of two tests. The testing was completed with 46 children with CI.

In regular schools, 200 research information and general data questionnaires were distributed, and signed consent was obtained for 118 children with NH (59%). A total of 110 hearing children participated in the testing, while 8 children for whom consent had been obtained were not tested, due to non-attendance at school, incomplete data or withdrawal. All hearing children underwent an individual hearing screening and met the inclusion and exclusion criteria.

The examined groups were equal in relation to age, sex and place of residence (Table 1). All subjects were of Caucasian (white) race. Descriptive analysis of socio-demographic and socio-economic characteristics of parents and family included data on age, professional training, professional status and parental marital status. The respondents were equal in relation to all comparable analyzed variables (Table A1).

All children from NH went to public, or regular schools and followed the regular educational plan and program. In contrast, children with CI attended either regular schools or schools for the education of children with developmental disabilities. The distribution of children with CI in relation to the type of school and level of educational support is presented in Table 1.

The characteristics of children with CI in relation to deafness and implantation of the CI were obtained from medical records, checking and possibly supplementing the data with the help of parents/guardians. The data referred to the age at the time of onset and etiology of deafness, hearing threshold before CI, age of the child at the beginning of rehabilitation and at CI implantation, date of implantation, type and model of CI, type of hearing amplification, and mode of communication. Data on the children’s psychological status, i.e., whether their non-verbal intellectual abilities were within average values, were obtained from the documentation.

In 32.6% of children with CI, the deafness was genetic, and in 37% it was of non-genetic origin. For about 30% of respondents with CI, the etiology of hearing impairment was unknown (Table A2).

The average hearing threshold in the left ear before CI was 99.2 ± 3.19 dB, and in the right 98.56 ± 3.78 dB. All children, except for two (4.36%) whose deafness was due to meningitis, used a hearing aid for at least three months prior to implantation.

In the group of children with CI, 30 (65.2%) had a Nucleus 24 cochlear implant, 15 (32.6%) a Med-El, and only one subject (2.2%) a Bionics cochlear implant. All subjects were unilaterally implanted. In 24 subjects (52.2%), the CI was implanted in the right ear, and in 22 (47.8%) in the left ear. Only 3 respondents (6.5%) used bimodal amplification, which, in addition to the CI, also involved the use of a conventional behind-the-ear hearing aid on the non-implanted ear.

The average age of subjects with CI at the beginning of rehabilitation was 23.6 months (Table 2). On average, subjects with CI were 43.15 months old at the time of implantation. Only two subjects were implanted in the second year of life, before the age of 24 months. There were 14 subjects in the group of children who were implanted in the third year of life. Ten subjects were implanted in the fourth year, and the largest number in the fifth year of life. The time period of the use of the CI ranged from 5 years to 13.5 years.

Not a single child with a CI used exclusively gestures or sign language in communication; 35 children with CI (76.1%) communicated exclusively orally, while 11 children (23.9%) used total communication, which implies the use of sign language and fingerspelling in addition to oral speech. None of them were native signers. They used local variants of Serbian Sign Language and a one-handed or two-handed finger alphabet.

In the group of children with CI, 10 of them (21.7%) had deaf/ hard of hearing parents or siblings in their immediate family. In the control group, one respondent had a close family member who was deaf or hard of hearing.

The testing of children was conducted individually, in a separate room or classroom, with an appropriate noise level, in order to be able to perform hearing screening and conduct testing.

Checking the condition of hearing in hearing children was performed using transiently evoked otoacoustic emissions [47]. With this simple, non-invasive, objective technique, data on the functioning of the cochlea is obtained, whereby a regular finding (“passed”) speaks in favor of preserved hearing up to 35 dBnHL in the hearing range from 1000 to 4000 Hz. Testing was performed using the AccuScreen TE miniature automated otoacoustic emission device, Otometrics (Madsen [48]). The procedure itself consisted of placing a probe with a speaker and a sensitive microphone in the subject’s external auditory canal, playing a signal and recording the response. The test lasted on average about 30 s for each ear, and required the absence of noise [49]. Data on the outcome of the procedure are recorded in the general questionnaire.

The total time required to conduct the testing was about 15 min per child. The same order of testing was applied with all children (first verbal fluency—phonemic and semantic, then figural fluency and finally arithmetic fluency).

### 2.2. Instruments

The verbal fluency test assesses the ability to create strategies in the verbal domain and is a version of the Controlled Oral Word Association Task, known as COWAT [50]. Phonological fluency was assessed using the sounds /S/, /K/, and /L/. The test taker is asked to enumerate as many nouns as possible in a given sound in one minute, while respecting certain rules (no proper nouns, toponyms, numbers, same words with different suffixes and words that do not belong to the requested group). During the instruction, the examiner showed the children with Cis the appropriate sound with a finger alphabet sign. During the implementation of the test, an auditory recording was made for each respondent, and the words produced by the respondents were recorded on a form. Based on the answers, the number of correct words, the number of repeated words, the number of rule violations and the number of illogicalities were calculated. The productivity score (the total number of correct words in all three sounds), and the percentage of errors in relation to the total number of produced words were used as the basic variables.

In this study, semantic fluency was assessed using the Abbreviated Categorical Naming Test [51]. The respondent is given the task of listing as many words as possible from the category of animals within 60 s, without repetition, proper names or naming animals of different age and gender if they belong to the same species. The category of animals was chosen because it is familiar to subjects with hearing loss, and it has been used in other studies [52,53,54]. The principle of recording answers is the same as in the previous test.

Figural fluency was tested using the Five-Point Test [55]. This test measures the ability to plan and strategize, divergent thinking and the ability to mentally “shift”. The test takers are asked to draw as many different figures as possible on A4 paper with 40 squares each containing 5 symmetrically distributed dots, connecting the dots in the squares with straight lines, where not all dots have to be used. The figures should not repeat, and lines which do not connect the dots should not be made. Prior to the actual testing, the examiner gave oral instructions and rules, and demonstrated two possible examples on a separate piece of paper, after which the respondent began to perform the task. The test took 2 min. The number of well-formed figures, the number of perseverative errors and the number of figures in which the lines do not connect the dots are counted. The number of errors is converted into a percentage value in the further analysis, by dividing the number of errors by the number of well-formed figures and multiplying by 100 [56]. The test showed a high interrater correlation (r = 0.99), a test-retest correlation of 0.77, as well as a high correlation with other fluency assessment tests (r ≥ 0.50) [57].

Arithmetic fluency measures the ability to quickly solve simple tasks involving addition, subtraction, multiplication and division up to 100 in a given period of time. For the purposes of this research, a test based on other arithmetic fluency tests was used. The tasks were presented on a laptop, using a Power Point presentation. Each task was presented separately, on a white background, in Calibri (body) font size 54 points. In the test that was applied in this research, the tasks had one or two operations, type 4 + 5; 9 − 5; 9 × 3; 90:15; 62 + 7 − 15. There were 120 tasks in total, with gradually increasing complexity of the tasks. In a pilot study, which was conducted with 20 children from the general population, the test showed high reliability, with the test-retest being 0.88. The respondent is instructed to solve the tasks as quickly as possible and write down on paper the results of the tasks that he sees` on the screen, and to move on to the next task when he finishes the task by pressing the key on the keyboard (space bar). If the respondent is unable to solve the task, he can move on to the next one, marking “-” on the paper for that task. Prior to the examination itself, it was checked whether the respondent understood the instructions, using three examples. The time was limited to 3 min. For further analysis, the number of correctly completed tasks and the number of incorrectly completed tasks were used. Since the respondents solved a different number of tasks on the tests and the possibilities for the occurrence of errors were not uniform, the errors are shown through the proportion of the number of errors in relation to the total number of solved tasks.

### 2.3. Statistical Data Processing

Data analysis was performed using the SPSS 19 for Windows statistical package. The descriptive part of the analysis shows absolute numbers, percentages, arithmetic mean (AS), range of values—minimum and maximum, and standard deviation (SD). Prior to further analysis, the normality of data distribution on continuous variables was checked using the Kolmogorov–Smirnov test. Since none of the variables had a normal distribution (Table A3), the Mann–Whitney U-test was used to compare the examined and control groups and within the groups. The relationship between the examined variables and the data related to the cochlear implant was checked using the Spearman correlation coefficient and partial correlation. The significance level was set at a *p* value less than 0.05. The Bonferroni correction was used for the correction of *p*-values in multiple correlations.

## 3. Results

Average scores on the fluency tests are presented in Figure 1, Figure 2 and Figure 3. The comparison of the groups was performed using the Mann–Whitney test. Differences in the percentage of errors on the fluency tests between the studied groups are shown in Table 3.

The Mann–Whitney test was used to analyze differences in relation to the sex of the respondents. In the group of children with CI, a statistically significant difference was obtained only in relation to phonemic fluency, where girls produced on average 19.36 ± 7.30 correct words, and boys 15.33 ± 7.65 (*Z* = −2.177; *p* = 0.029). In the group of subjects with preserved hearing, no statistically significant difference by sex was obtained on any test.

In order to ascertain the correlation of fluency test results with age, sex of respondents with CI and characteristics related to CI, Spearman’s correlation was used, and the results are shown in Table 4.

The age of respondents with CI was significantly positively correlated with PF and strongly positively correlated with AF, while sex was moderately correlated only with PF. With further analysis, using the Bonferroni correction of *p*-values, the correlation between age and gender on the one hand, and PF on the other, was no longer statistically significant. Due to the extremely high correlation between chronological age and hearing age (rho = 0.835; *p* < 0.001), a partial correlation was performed, whereby the age of the subjects with CI was controlled. Following the correction, only SF was significantly positively associated with children’s chronological age, while the association with PF and AF was lost (Table 4). When the Bonferroni correction is taken into consideration, statistical significance between hearing age and SF is also lost.

The analysis of the relationship between the results on the fluency tests was performed using Spearman’s rho coefficient, and the obtained values are presented in Table 5.

A statistically significant positive correlation was obtained between all examined types of fluency, in both groups. The strongest relationship was obtained between FF and AF in children with CI. The statistical significance remained after applying the *p*-value correction using the Bonferroni method.

## 4. Discussion

Cochlear implantation has been applied in the Republic of Serbia since 2002 [58]. Children who were among the first to be implanted were included in the research, and were between the ages of 9 and 16 years at the time of the research. The long-term use of CI by our respondents gives the opportunity to observe the long-term outcomes of CI in different domains of life. Unlike the majority of research in our environment, but also more widely, this research did not focus on speech-language development and listening abilities, but rather on cognitive functioning in terms of verbal and non-verbal fluency, as well as arithmetic fluency. The results of our research showed that the achievements of children with CI on all applied fluency tests were significantly poorer compared to their hearing peers, and a correlation between the other studied fluency modalities and arithmetic fluency was also obtained.

As expected, children with CI produced a significantly lower number of words on verbal fluency tasks compared to their hearing peers. A key feature of verbal fluency tasks is that they cannot be performed using standard, stereotyped, learned programs, but rather require the respondents to establish and implement their own search strategy and retrieve content from semantic memory in order to produce words in a specific sound or from a specific category [59]. Saturation with the verbal factor, both in assigning and performing tasks on verbal fluency tests, contributes to these results of children with CI. Although respondents had significantly different achievements on both tests, the smaller difference between groups was on the SF test, where the number of errors that respondents made during this task did not differ between groups. The range of responses within SF in both groups is similar. It can be assumed that the category of animals, which was used in the SF test, is well known to children with CI, as through rehabilitation and education, children with CI encounter concepts from this category almost daily. Reduced production in children with CI can be explained by a less developed spoken-language vocabulary [53,60,61], a different organization of semantic memory [54], and a poorer ability to create adequate strategies for memory search [52,62]. Other researchers [53,54] report similar data for children with CI, obtained using the same test, but there are also studies that did not find a difference between the number of words produced on the SF test [63]. A study that compared deaf adults who use American Sign Language and fingerspelling and hearing people, on tasks of semantic fluency, showed that the results of these two groups are similar, especially in a case of native signers [64]. In this study, children with CI, although some used sign language and fingerspelling in their communication, were not native signers, which could have contributed to the poorer results. Kenneth et al. [52], analyzing the semantic network using the SF task (in the animal category), report a lower degree of development of the structural semantic network in children with CI compared to their hearing peers, aged 7 to 10 years. Socher et al. [65] also obtained similar results regarding the semantic network and the number of words, with this difference disappearing when considering the hearing age of the respondents. The connection between SF and pragmatic abilities of children with CI is interesting, and was not found in children without hearing impairment [65]. In the study by Da Giacomo et al. [66], VF was used as a measure of cognitive flexibility. Children with CI (*n* = 17), with a mean age of 8.8 years, produced an average of 14 words on the PF test and 34 words on the SF test, and in both cases, this was significantly fewer compared to a group of hearing peers. A high correlation with working memory capacity was obtained. Respondents with CI in this study produced a slightly higher number of words compared to the aforementioned research, which can be explained by their older chronological age. In addition, there was a positive correlation between SF and the hearing age of children with CI, which means that over time, by enriching experiences and vocabulary, the abilities to recall and produce adequate words from a certain semantic category increase. In our research, there was no correlation of semantic fluency with sex, age of implantation/activation of CI, or the beginning of rehabilitation.

Phonemic fluency is a more difficult task than SF, given that words belonging to a certain semantic category are lexically connected in a larger coherent group, which facilitates their recall [67]. PF tasks require respondents to use certain search strategies that are not habitual and targeted, and require a higher level of organized search and inhibitory control. The achievements of children with CI in our study were poorer than those of children with NH, both in terms of the number of correctly produced words (nouns) and the number of errors. Difficulties in the achievement of children with CI in producing words on this test can be partially explained by the deficit that is generated from the phonological, through the morphological to the semantic level of language. Phonological processing and representation, as well as the ability to analyze phonemes, are at the core of the PF task and allow the subject to generate words that begin with a specific sound (phoneme). The sensitive period for the beginning of the development of phonological abilities for the sounds of the native language is the second half of the first year of life [68], which means that in children with congenital deafness and hearing loss in the first months of life, these abilities begin to develop only after the implementation of the CI and in conditions of limited auditory information [69]. According to Pisoni et al. [70], the degraded signal that reaches the auditory cortex in children with CI further complicates the development of phonological presentation in long-term memory, which remains insufficiently specific and precise. Consequently, other systems responsible for higher levels of language processing such as recognition, categorization, lexical discrimination and selection, which rely on verbal auditory processing, also do not reach an adequate level of development [71,72]. It is considered that the morphological structure of words reflects phonological processes, which are often deficient in children with CI [72]. The morphological level overlaps with the phonological level, but the awareness of the structure, i.e., the root basis of the word, develops much later. In the lower grades of elementary school, teachers draw attention to words that have a common basis (e.g., sun-sunscreen; hear-hearing; horse-horseback). In the upper grades of elementary school, the concept of word roots and phonetic changes are taught, which often in our language can “blur” the same word origin (e.g., chicken-chick; dog-dogs; book-bookstore). Deaf and hard-of-hearing children struggle with recognizing the common basis of these words and they remember these words as separate, unrelated ones [73]. Difficulties in this segment of processing (morphological level) lead to rule-breaking errors in PF tasks, although the possibility that children, who understand the concept of the root base of a word, forget the requirement to follow the set rule cannot be ruled out. As previously mentioned, at the lexical level, the spoken-language vocabulary of deaf and hard-of-hearing children is significantly smaller than that of hearing children [53,60,61]. Unlike hearing children, who acquire most new words spontaneously, by casual learning in their immediate environment, listening to the speech of their family and friends, and through the radio or TV, deaf and hard of hearing children learn almost every word by being taught by a teacher, therapist or parents [74]. In addition, if the word is not adequately practiced, it is unlikely that the child will spontaneously remember it. In addition, it is known that the thinking of deaf children who are not exposed to early language stimulation may remain at the concrete level for a long time, thereby making it more difficult to acquire abstract nouns [75]. During the testing, this was manifested in the fact that children with CI, after receiving a stimulus—a phoneme, observed objects around them, looking for those that begin with the given sound, and later attempted to create certain mental strategies for searching words in long-term memory. Although this also happened in children with NH, they still switched to the internal verbal deposit more quickly and produced a greater number of words. Due to all of the above, the poorer achievements of children with CI on the PF examination test can be explained by a lag in language development (on the phonological, morphological and semantic levels), a concrete way of thinking and insufficient development of the speed and fluency of processing verbal information, i.e., the use of adequate recall strategies of words from long-term memory with simultaneous rule-following and inhibitory control.

In this study, girls with CI produced a higher number of words on the PF test. A meta-analysis by Hirnstein et al. [76] supports a PF-level advantage in favor of girls in the general population over the life span. The data show differences in the area of phonological development and phonological awareness in favor of girls [77]. Cupples et al. [78] found no gender differences between children with hearing loss on phonological awareness tasks at ages four and five, while Geer [79] found better achievement in reading skills in girls with CI, aged eight to ten. Furthermore, in other studies of children with CI, girls achieved better results in the area of language development, such as grammar [80] and expressive language [81]; however, we were unable to find data on sex differences on PF tests. Considering the varying results regarding phonological awareness in the general population and in CI users, further research is required to shed light on whether differences are present at all ages, i.e., if and when girls have an advantage on PF tasks, but also on other aspects of phonology awareness.

Children with CI also achieved poorer results on the test which measures figural fluency, compared to children with NH, in this research. This result, along with a significantly higher average percentage of errors (doubled) by children with CI, speaks in favor of slower and more difficult processing, i.e., lesser ability to generate non-verbal responses and their control. In contrast to our results, Figueras et al. [82] did not find a difference between respondents with CI, hearing aids and hearing peers using the Design fluency test from the NEPSY battery in children aged 8 to 12 years. However, the number of correct figures in respondents with CI was lower compared to the other groups. Other authors, whose results align with ours, consider that the weaker achievements of deaf children on non-verbal tests such as the FF and similar tests are related to their conceptual concreteness and rigidity [57], planning skills and difficulties in the field of visual spatial abilities [83,84]. The Five-point Test [55] requires the respondent to create new figures on his or her own, and there are a large number of different correct solutions or answers. For this reason, it is considered that tests of different types of fluency also represent a measure of divergent thinking, and poorer results could indicate difficulties in this area of cognitive functioning as well. Taking into account the achievements on the test (Five-point Test), it can be assumed that children with CI have difficulties with the development of strategies which enable the creation of new original figures and prevent the repetition of previously drawn figures. In the research of Boerrigter et al. [83], a wide set of language abilities and executive functions were examined in children with CI, HA and hearing peers, with an average age of 11.85 years. There were no differences between the groups on the Design Fluency test, but there were on the test which examined planning. As figural fluency tests also require planning, it can be assumed that part of our results could be explained by planning difficulties. Marschark et al. [84], comparing the visuo-spatial abilities of deaf/hard of hearing students (with and without CI) and hearing first-year college students, found that deaf/hard of hearing students achieve worse results, and that there is a possibility that their visuo-spatial abilities are related to different cognitive processes. Other authors report an advantage in the domain of visuo-spatial abilities, such as visuo-spatial perspective, and mental rotation tasks in deaf people who are native signers [85,86,87]. In our research, even the children with CI who used sign language in their communication were not native signers, so the obtained results of figural fluency cannot be generalized. The FF test is not completely free of the verbal factor either. Although the respondent is shown what is expected of him or her prior to testing, part of the instructions and orders are presented verbally. Variables related to the age and sex of the subjects, the age at the beginning of rehabilitation and at the time of CI implementation and hearing age were not related to the number of well-formed figures.

Conway [88] considers that poorer results on most tests for assessing EF in children with CI are precisely the consequence of a deficit in processing speed and fluency. Pisoni [89] also suggested that processing efficiency and speed might have a general effect on EF. According to this author, in children with CI, the problem is at the level of basic information processing skills, which are manifested through the efficiency of presentation and the capacity to process information, regardless of whether it is verbal or non-verbal. Some other research with children with CI, also speaks in favor of a higher risk for the weaker development of EF or in some EF domains [24,90]. Another group of authors speaks in support of the importance of early exposure to sign language and its protective role for the development of EF in deaf children [12]. By functional assessment of EF, using questionnaires filled out by parents, they determined that deaf children who were native signers, did not differ from their hearing peers in any of the examined domains of EF. The mentioned research supports the hypothesis of language deprivation as a possible cause of difficulties of deaf and hard of hearing children in the field of EF, information processing, and neurocognitive development. Research by Chang et al. [91] using neuroimaging techniques, confirms the importance of early exposure to sign language in deaf people for neurocognitive development in areas responsible for language processing. In our sample, there is practically a combined effect of early auditory and language deprivation, because subjects with CI were not exposed to early linguistic stimulation through sign language, and if some of them had some auditory experiences, they were short-lived or incomplete. Considering the above, supporting early exposure and acquisition of sign language in deaf children, regardless of whether cochlear implantation is planned or not, could be a protective factor for their cognitive development.

As previously mentioned, in the arithmetic fluency test, in this research, significantly poorer achievements of children with CI were found compared to their hearing peers. Arithmetic fluency test scores reflect the level of balance established between conceptual, procedural, and declarative knowledge [92]. Although efficiency on tests of arithmetic fluency can be achieved even without comprehension, conceptual knowledge facilitates the completion of tasks and is of great importance in adapting and applying procedures to new, unfamiliar tasks and problems [93].

According to Menon [94], the basis of arithmetic skills is the sense of number, which includes quantity and the principle of cardinality, as well as manipulation of numerical quantities. This first, basic level of processing requires the integration of the visual and auditory cortex, which enables the decoding of visual forms and phonological characteristics of stimuli, and the parietal attention system, whose role is reflected in the semantic representation of quantity. The next system represents cognitive processing and working memory, based in the basal ganglia and frontoparietal network, which, by creating short-term representations, support the manipulation of multiple discrete quantities (and processes). The system of episodic and semantic memory, as the third link, plays a significant role in the creation of long-term memory and generalization. The fourth system, prefrontal control processing, is responsible for directing and maintaining attention for the purpose of making goal-directed responses. Children with CI, who have lost their hearing prelingually, and were not exposed to early language acquisition, regardless of modality, may have difficulties already at the first, basic level, which could later make further processing difficult.

In addition, there is a possibility that children with CI in our study organize, store and recall facts from long-term memory in a different way, that is, that they rely more on non-verbal visuo-spatial components when solving tasks quickly. Cragg et al. [95] consider that arithmetic facts are stored in long-term memory in verbal form, but that they inevitably contain a visuo-spatial component, considering the way sums are usually presented. Based on this, it can be assumed that children with CI rely more on visual presentation of facts than on verbal presentation on AF tasks. On the one hand, greater reliance on visuo-spatial sketches within the framework of solving arithmetic tasks represents a strategy characteristic of younger children [96,97], while reliance on phonological, verbal information within working and long-term memory represents a more advanced strategy [98,99]. On the other hand, storing and recalling arithmetic facts in children with CI is difficult, due to the verbal nature of the information, and therefore, children are more inclined to apply a procedural strategy, i.e., they solve tasks as new each time, while children with preserved hearing use, for example, memorized tables for multiplication and division. This is reflected in the slower completion of tasks and the greater number of mistakes made by children with CI. This method of calculation is found in children who have below-average achievements in solving arithmetic problems [100]. In a study in Sweden, no differences were found between deaf signers and hearing non-signers in subtraction tasks, while deaf signers had worse achievements in multiplication tasks. The authors explain the obtained results by the reliance of deaf signers on the phonology of sign language and the incomplete automatization of simple arithmetic facts [101]. The results obtained by our research indicate the possibility of using a visual presentation of facts in long-term memory, less well-established automatization of arithmetical facts, and/or applying a less advanced strategy by children with CI that would be expected in deaf children who use sign language [102]. Frostad [102] pointed out that although sign language can make it difficult for deaf children to acquire conceptual knowledge related to numbers, it provides them with additional strategies that facilitate procedural efficiency.

The age of the subjects was strongly positively related to the number of correctly completed tasks within arithmetic fluency, while sex, age at the beginning of rehabilitation and at the time of implant placement were not. Although hearing age was also associated with AF, when controlling for age, it was no longer statistically significant. In a longitudinal study with children with CI, Thoutenhoofd [10] found that their arithmetic skills are related to age, and that the difference compared to hearing peers decreases with increasing age.

The possible existence of difficulties in the area of calculation and mathematical thinking already at the beginning of formal education, as well as their predictive value in later achievements in mathematics, speak in favor of the need for early identification of these difficulties and timely intervention, in order to prevent difficulties from accumulating [36,103]. Langdon et al. [104], analyzing a large number of studies on the acquisition of early numerical concepts in deaf children, state that in this area as well, early exposure and acquisition of language has proven to be important and a protective factor, regardless of language modality. In addition to language, the authors point out that a stimulating environment, with “mathematical” talk and plenty of opportunities to observe and understand visuo-spatial representations, is the best way to build a foundation for future more complex mathematical knowledge and skills in young deaf children, constituting “key goals for early intervention programs” [104].

Although not examined in this research, some of the possible factors that could be related to the achievement of students with CI on all fluency tests could be the type of school, school program and additional educational support. At the time of conducting the research, two educational systems overlapped in Serbia—the old one with special education and the new one with inclusive education, which was introduced in 2009, so some children with CI attended classes according to the old system and some according to the new. The decision about which school the children will attend depended (and currently depends) only on the parents who, based on the child’s and their own needs and possibilities, decide whether the child will go to a school for children with developmental disabilities or a regular school. In many regular schools, support for deaf and hard-of-hearing students has been largely nonexistent, and it cannot be determined exactly what accommodations and modifications have been made. On the other hand, the special program and special schools, although they could be more restrictive than the regular ones, provided children with additional support and programs for further development of listening, speech and language and support in learning.

Considering the assumption that different forms of fluency have a common cognitive basis and processes, numerous studies have examined their connection [44]. Balhinez and Shaul [44] believe that in order to improve mathematical fluency and reading fluency, it is necessary to focus on strengthening the underlying cognitive abilities, and according to their research, these are fluency, working memory and inhibition. Phonemic and figural fluency are predictors of mathematical achievement in children with normal hearing [105]. Although AF represents a narrower view of mathematical skills, there is agreement among authors that it is a predictor of general mathematical achievement, either in hearing children or in deaf and hard of hearing children [45,106,107]. Verbal fluency has been shown to be highly correlated with the vocabulary of children with CI, and receptive and expressive vocabulary are some of the strongest predictors of reading and writing ability [15,53,108]. Therefore, there is a need to further examine their relationships and possible impact on the academic skills of children with CI, such as writing, reading and arithmetic skills.

In our study, PF and SF were similarly related in both children with CI and hearing children. A strong correlation (r = 0.48) between PF and SF in children with typical development in the third grade of elementary school was also obtained by Aksamovic et al. [109]. In the research of Kavé and Sapir-Yogev [110], among other things, the connection between PF and SF through the life-span was analyzed. At the age of five to seven years, the correlation coefficient was 0.16 and from 8 to 10 it was 0.40; the strongest correlation was obtained in subjects aged 11 to 13 years, at 0.64. Although many studies have examined PF and SF in children with CI, there is no analysis of the relationship between them. They are usually linked to other abilities/skills (e.g., reading or pragmatic abilities) or a composite score is created [15], so our results cannot be interpreted in relation to other research with children with CI. However, it can be concluded that the relationship between PF and SF is similar to that of children with typical development, both in this and other research.

In this study, figural fluency was slightly more highly related to PF and SF in the group of children with CI, compared to the control group. A strong correlation was present in children with CI, while it was moderate in relation to PF and weak for SF in children with NH. In the general population, authors generally report a low association between FF and VF, with the correlation between SF and FF ranging from 0.21 to 0.37, and that between PF and FF ranging from 0.22 to 0.38 [111,112], while in some papers no association was obtained [113]. In a study by Marshall et al. [53] in deaf children aged 6 to 11 years, a correlation was obtained between semantic fluency and the Design Fluency Test (r = 0.38). Although these results are comparable to ours, the stronger correlation in our study could be attributed to a higher average age, the use of CI, or the use of a different test.

In the group of children with CI, AF was strongly positively associated with SF and FF, while the association with PF was borderline significant. Even in children with preserved hearing, the association was strong. Research in the general population supports these results, where the correlation coefficient is around 0.4 [114]. As of yet, we have not come across data on the connection between arithmetic, verbal and non-verbal fluency in the literature. However, researchers have examined the relationship of mathematics to more complex concepts, such as reading, intelligence, etc. Huber et al. [115], examining the arithmetic skills of children with CI, obtained results different from ours; the achievements of children with CI were at the level of their hearing peers, and reading skills were singled out as a predictor, while non-verbal intelligence was not related to achievements in arithmetic. In contrast, in refer to hearing children, nonverbal IQ was positively associated with arithmetic achievement, while reading skill was not. Other authors emphasize the importance of language, that is, phonological representations and processes in manipulating and storing verbal codes, such as counting and solving simple arithmetic tasks [116,117]. Phonological awareness most likely affects arithmetic skills due to the high-quality phonological representation of some aspects of arithmetic, such as fact recall [118].

There are several limitations of the study. First of all, the research was organized as a cross-sectional study, which enabled immediate insight into the examined functions. Considering their developmental nature, a more complete picture, as well as a basis for guidelines in working with children with CI, could be obtained by long-term monitoring and examination of other cognitive functions, such as working memory, inhibition and mental shifting, which were found to be significant in other studies. In addition, non-verbal intellectual abilities, which represented criteria for inclusion in the study, were obtained from clinical or school documentation, but the focus was only on whether the ability was within the normal range; hence, it cannot be claimed that the examined groups were equal in non-verbal intellectual abilities. Although all children with CI participated in audio-verbal therapy, the length and specifics of the therapy were not taken into account. As in many other studies, it is very difficult to achieve homogeneity in the group of respondents with CI. Children who were born deaf or lost their hearing during the first two years of life were included in the sample. There is a possibility that the early auditory experiences of children who were born hearing could have influenced the results. Likewise, the additional heterogeneity in the group of respondents with CI is a consequence of the wider range within the time frame of CI implantation. Our desire was to include children who were among the first to be implanted, who so far had not been monitored in other areas of functioning, except for speech and language. Furthermore, one of the criteria was the absence of additional disorders in children with CI, in order to obtain as homogeneous a group of respondents as possible. In real life, the number of CI users with multiple disabilities is increasing, so future research must also take them into account.

## 5. Conclusions

There is very little data in the literature about the arithmetic skills of children with CI, and this is the first study in our country, and one of the rare studies in the area and beyond. The obtained data suggest the harmful effects of early auditory and language deprivation and later exposure to altered auditory experiences on a wider range of abilities than speech and language. Children with CI achieved significantly poorer results on verbal and figural fluency tests compared to their hearing peers. In addition, children with CI performed more poorly on the arithmetic fluency test. The significant connection between the observed modalities and types of fluency speaks in favor of a common basis and the sharing of certain cognitive processes. Further examination of various aspects of cognitive abilities and mathematical skills, as well as the factors associated with them, are necessary in order observe the specifics and needs of children with CI, and to act preventively in good time or apply a timely intervention which would remove or mitigate the limiting effect of the deficiency or incomplete listening.

## Figures and Tables

**Figure 1 behavsci-13-00349-f001:**
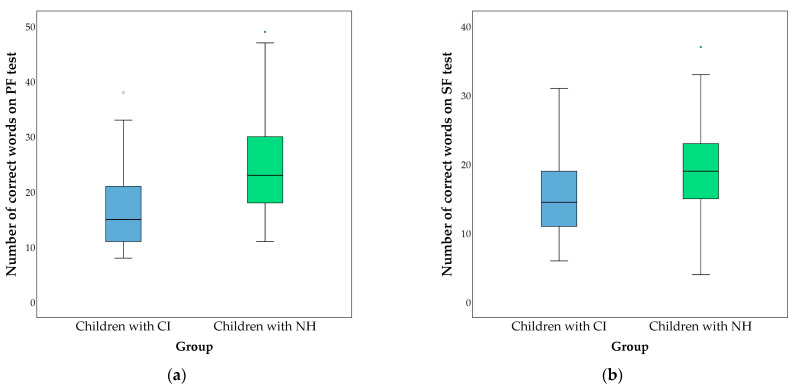
(**a**) Comparison of numbers of correct words on the phonemic fluency (PF) test between children with CI (represented in blue) and children with NH (represented in green). Children with CI had 17.521 ± 7.653 correct words; children with NH had 24.927 ± 8.784. A statistical difference was found (*Z* = −4.916; *p* < 0.001). (**b**) Comparison of numbers of correct words on the semantic fluency (SF) test between children with CI and children with NH. Children with CI produced 15.044 ± 5.428 correct words; children with NH had 18.855 ± 5.474. A significant difference was found (*Z* = −3.889; *p* < 0.001).

**Figure 2 behavsci-13-00349-f002:**
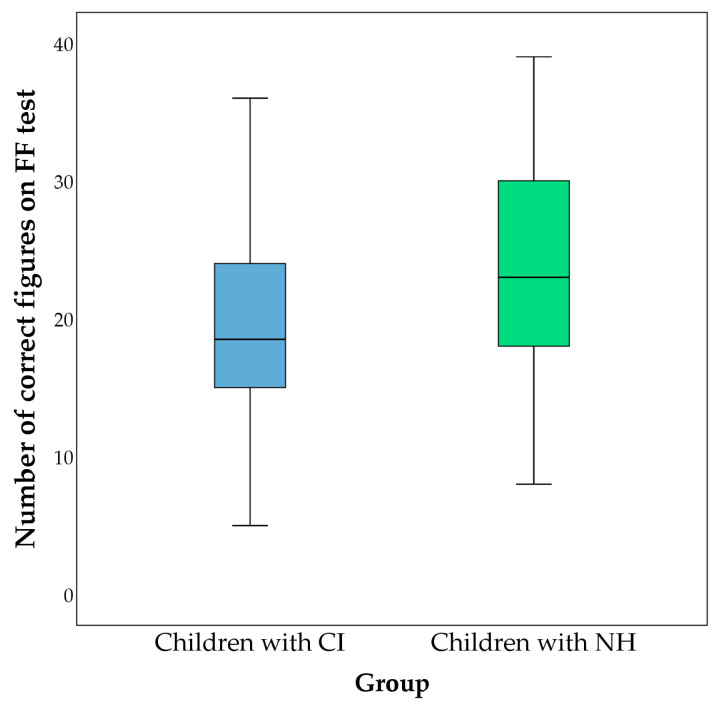
Comparison of numbers of correct figures on figural fluency (FF) test between children with CI (represented in blue) and children with NH (represented in green). Average numbers of correct figures in the group of children with CI were 18.957 ± 6.706, and 23.373 ± 7.711 for children with NH. A significant difference was found (*Z* = −3.068; *p* = 0.002).

**Figure 3 behavsci-13-00349-f003:**
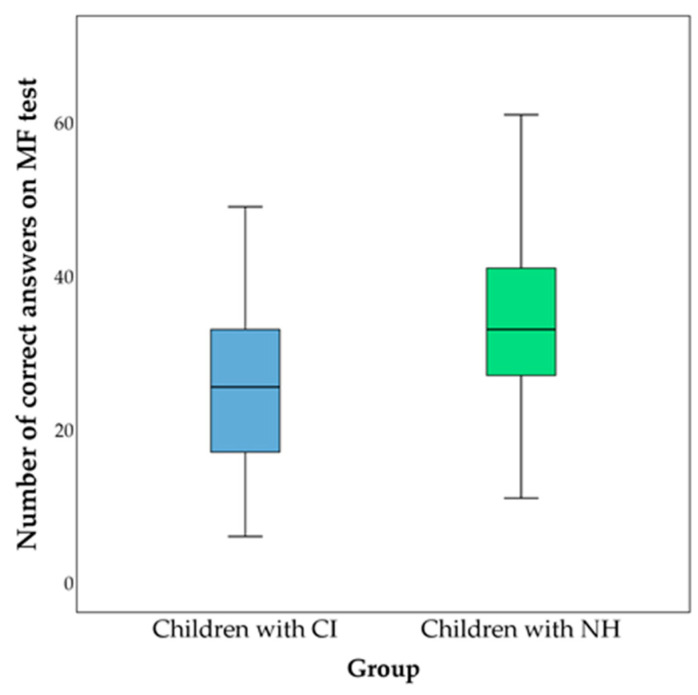
Comparison of numbers of correctly completed tasks on an arithmetic fluency (AF) test between children with CI (represented in blue) and children with NH (represented in green). Children with CI had 25.870 ± 10.269 correct answers; children with NH had 34.000 ± 9.280. A significant difference was found (*Z* = −4.273; *p* < 0.001).

**Table 1 behavsci-13-00349-t001:** Socio-demographic characteristics of children with CI and children with NH.

	Children with CI	Children with NH	Test Statistics
Age (years; months)			
Minimum-maximum	9;0–16;0	9;3–15;11	*t* = 0.802; *p* = 0.424
AS ± SD	12;11 ± 2;1	12;9 ± 1;10	
Sex *n* (%)			
male	21 (45.7%)	50 (45.5%)	*χ*^2^ = 0.001; *p* = 0.982
female	25 (54.3%)	60 (54.5%)	
School setting *n* (%)			
Regular school	27 (58.7%)	110 (100%)	NA
School for children with disability	19 (41.3%)	-	
Educational curriculum *n* (%)			
Regular	15 (32.6%)	110 (100%)	NA
Regular with accommodations	11 (23.9%)	-	
IEP/special curriculum	20 (43.5%)	-	
Living area *n* (%)			
urban	21 (45.7%)	46 (41.8%)	*χ*^2^ = 0.240; *p* = 0.887
suburban	6 (13%)	12 (12.7%)	
rural	19 (43%)	50 (45.5%)	

Note: CI—cochlear implant; NH—normal hearing; IEP—individualized educational plan; NA—not applicative.

**Table 2 behavsci-13-00349-t002:** Age of children with CI expressed in months, at the beginning of rehabilitation, at the time of implantation of CI and length of use of CI.

	*n*	Minimum–Maximum	AS ± SD
Beginning of the rehabilitation	46	7–48	23.587 ± 10.215
Age at implantation	46	21–65	43.457 ± 13.106
Hearing age	46	60–163	110.717 ± 21.706

**Table 3 behavsci-13-00349-t003:** Comparison of errors percentages on the fluency tests between children with CI and children with NH.

	Group	Minimum–Maximum	Mean ± SD	*Z*	*p*
Phonemic fluency	CI	0–52.38	8.661 ± 10.297	−3.650	<0.001
	NH	0–26.32	3.785 ± 4.923		
Semantic fluency	CI	0–14.29	2.344 ± 4.048	−1.108	0.268
	NH	0–14.81	1.564 ± 3.348		
Figural fluency	CI	0–46.67	10.661 ± 12.302	−2.160	0.031
	NH	0–33.33	5.201 ± 5.392		
Arithmetic fluency	CI	0–71.43	10.725 ± 13.911	−1.454	0.146
	NH	0–33.33	6.443 ± 6.690		

**Table 4 behavsci-13-00349-t004:** Correlation between age, gender and CI characteristics and correct answers on the fluency tests in children with CI.

	Phonemic Fluency	Semantic Fluency	Figural Fluency	Arithmetic Fluency
Age	0.359 *	0.128	0.228	0.531 ***/^†^
Sex	0.325 *	−0.077	0.086	0.030
Beginning of rehabilitation	0.026	−0.195	−0.041	−0.061
Age at implantation	0.113	−0.192	0.027	0.075
Hearing age	0.352 *	0.265	0.199	0.624 ***/^†^
Hearing age controlled by chronological age	0.043	0.311 *	−0.010	0.271

* *p* ≤ 0.05; *** *p* ≤ 0.001; ^†^ Correlations which remained significant after the Bonferroni correction (*p* ≤ 0.005).

**Table 5 behavsci-13-00349-t005:** Correlation between correct answers on the fluency tests in children with CI and children with NH.

	Phonemic Fluency	Semantic Fluency	Figural Fluency	Arithmetic Fluency
Phonemic fluency	–	0.485 ***^/†^	0.359 ***^/†^	0.507 ***^/†^
Semantic fluency	0.422 **^/†^	–	0.243 *^/†^	0.430 ***^/†^
Figural fluency	0.413 **^/†^	0.564 ***^/†^	–	0.458 ***^/†^
Arithmetic fluency	0.397 **^/†^	0.539 ***^/†^	0.659 **^/†^*	–

Note: Correlations for children with CI (*n* = 46) are to the left and below the diagonal. Correlations for children with normal hearing (*n* = 110) are to the right of and above the diagonal. * *p* ≤ 0.05; ** *p* ≤ 0.01; *** *p* ≤ 0.001; ^†^ Correlations which remained significant after the Bonferroni correction (*p* ≤ 0.013).

## Data Availability

The data presented in this study are available on request from the corresponding author.

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
