# Peer review of "Verbal, Figural, and Arithmetic Fluency of Children with Cochlear Implants"

_behavsci, 2023, doi:10.3390/bs13050349_

Round 1
Reviewer 1 Report
Dear Authors,
I want to congratulate the authors on this way of presenting their study.
However, several points as indicated below need to be addressed by authors to improve the quality of the article:
1. In the objective of the study, the authors speak of figurative fluency (lines 165-166), but in the title, abstract and throughout the introduction they refer to this same concept of non-verbal fluency. I suggest that you keep the same designation throughout the entire work.
2. The authors said: This cross-sectional study included 156 children, divided into two groups. The first group consisted of 46 children with a cochlear implant (CI), and the second group consisted of 110 children with preserved hearing (normal hearing - NH). (Lines 170-172). There will not be a very large difference in the number of subjects between the groups.
3. Were children with middle ear disorders included in the study, regardless of the group (experimental and control)? How did these cases work?
4. I appreciated the authors' care in controlling the sociodemographic, socioeconomic data and characteristics of the child related to the deafness variables, as they could have influenced the results.
5. The authors refer (lines 284-286) that: The productivity score (total number of correct words in the three sounds) and the percentage of errors in relation to the total number of words produced were used as a basic variable. Was the reaction time of the response not measured?
6. I appreciated the discussion is well reasoned.
7. The references used are very current Congratulations!
8. The conclusion can be more succinct and more result-oriented
Reviewer 2 Report
Thank you for the opportunity to review this article. It was a pleasure to read and will be a strong candidate for publication with the suggested revisions. Please find general and specific comments in the attached document.

Reviewer 3 Report
Dear Author, thank you for sending the paper and th idea to devote your research time and resources for children with CI This field still needfs a lot of work and i would like to encourage you warmly to continue your work in this field.
I would like, however, to direct your attention to the following issues:
1. Abstract: it is not the cochlear implantation that provides children with abilitis and skills. This is due to a long and complicated process of implantation and rehabilitation after.
2. lines 35-38 - the same issue - CI implantation cannot promise the issues mentioned - this is just one of many factors that influence development of the skills and competenctions mentioned
3. "the goals of today's cochlear im-40 plantation are speech intelligibility, without lip reading, using the phone and listening in 41 noisy conditions" - I suppose this is NOT the goals - the goals are much bigger - like e.g. harmonious integral development - these mentioned are just soem of the skills possible.
4. the achievements of children 49 with CI lag behind their hearing peers in the domain of cognitive and social functioning 50 7,8, and academic skills 6,9,10. - yes, some research might say so but there are other with opposite conclusions
5. line 55 - and many more conditions, precisely described in literature (like e.g. Archbold, Skarżyński et alters
6. line 59 - the same - the literatire review has not been done thoroughly so as to show the full picture of this issue
7. line 160: his question is not methodologically correct - there are 2 factors in one question (auditory deprivationa and listening analysed. What is more, there are many more factors that can condition children's fluenct and they have not been controlled in this study.
8. line 178 - this criterionis not precise: children with innate hearing loss who has neve had any language experienced form a different group from children who lost their hearing at the age e.g. 2
9. line 179 implanted cochlear implant before the age of 6 - this criterion is very wide and not correct - CI implantation at the age of 2 is a completely different situation than CI implantationa at age 5!
10. 182 - The control group consisted of children with preserved hearing, of average intellectual abilities, who successfully passed hearing screening (otoacoustic emissions) - the control group is twice as big as the research group. The criteria for inclusion into the research group are not precise. Such wide criteria are not methodologically correct to form a control group.
11. line 191 The research was conducted in the period between June 2016 and June 2017 - the researh comes from 2017 - it means 6 years ago and for CI technology it means a lot of changes,. If published this paper will be of "historic" value.
12. Another methodological problem: children with HL attended both mainstrteam and sepcial schools - this makes a difference of educational setting between the research and control group and is not correct. Such differnt groups cannot be compared!
13 CI children have a modified curriculum while children in control group attended regular program - such groups cannot be compared!
As mentioned, the study was not correctly designed and thus the results are not valid. Weaker results in verbal non-verbal and arithmetic fluenct migh tappear due to other factors, not CI itself.
With regret, I cannot recommend this paper for publication.

Round 2
Reviewer 3 Report
Dear Author/s, thank you for submitting the response to my review. It shows you worked heavily on the project - and on improving its presentation in the paper. I agree that different systems should be given a chance to be presented worldwide .
I generally accept your explanations and improvements. I suppose the texxt might be enriched also with some other literature avalable online - i suggest some good sources for the "verbal" part of youir research/
I am happy to recommend the text for publication. '
Domagała- Zyśk E. (red.), (2014). Developing language competence of people with hearing and speech disorders. Lublin: Wydawnictwo KUL.
Domagała-Zyśk E., Podlewska A. (2019). Strategies of oral communication of deaf and hard of hearing (D/HH) non-native users. European Journal of Special Needs Education, 34,2, 156-171. https://doi.org/10.1080/08856257.2019.1581399
Domagała-Zyśk E. (2021). To speak or not to speak? Speech and pronunciation of deaf and hard of hearing students learning English as a foreign language. In: Domagała-Zyśk E., Moritz N., Podlewska A. (ed.) English as a Foreign Language for Deaf and Hard of Hearing Learners Teaching Strategies and Interventions. London: Routledge, p. 17-31. DOI: 10.4324/9781003162179-3.
Domagała-Zyśk E. (2016). Vocabulary teaching strategies in EFL classes for deaf and hard of hearing students. W: E. Domagała-Zyśk, E.H. Kontra (red.) English as a foreign language for deaf and hard-of-hearing persons. Challenges and strategies. Newcastle upon Tyne: Cambridge Scholars Publishing, s. 135-152.
Author Response
Dear Reviewer
We would like to thank you for the time and effort in considering this manuscript. Your comments meant a lot to us and helped us to improve the manuscript.
Thanks for the recommended literature. We have used one of the mentioned paper in order to expand the discussion (lines 497-502).
Kind regards,
Authors